

# Prioritisation of structural variant calls in cancer genomes

Miika J. Ahdesmäki[1], Brad A. Chapman[2], Pablo Cingolani[3], Oliver Hofmann[4], Aleksandr Sidoruk[5,6], Zhongwu Lai[7], Gennadii Zakharov[5,8], Mikhail Rodichenko[5], Mikhail Alperovich[5], David Jenkins[9], T. Hedley Carr[1], Daniel Stetson[7], Brian Dougherty[7], J. Carl Barrett[7] and Justin H. Johnson[7]

[1] Innovative Medicines and Early Development, Oncology, AstraZeneca, Cambridge, United Kingdom
[2] Harvard T.H. Chan School of Public Health, Harvard University, Boston, MA, United States
[3] Kew Inc., Cambridge, MA, United States
[4] Centre for Cancer Research, University of Melbourne, Melbourne, Australia
[5] EPAM Systems Inc., Newtown, PA, United States
[6] Department of software engineering, St. Petersburg State University, St. Petersburg, Russia
[7] Innovative Medicines and Early Development, Oncology, AstraZeneca, Waltham, MA, United States
[8] Pavlov Institute of Physiology, Russian Academy of Sciences, St. Petersburg, Russia
[9] Boston University, Boston, MA, United States

## ABSTRACT

Sensitivity of short read DNA-sequencing for gene fusion detection is improving, but is hampered by the significant amount of noise composed of uninteresting or false positive hits in the data. In this paper we describe a tiered prioritisation approach to extract high impact gene fusion events from existing structural variant calls. Using cell line and patient DNA sequence data we improve the annotation and interpretation of structural variant calls to best highlight likely cancer driving fusions. We also considerably improve on the automated visualisation of the high impact structural variants to highlight the effects of the variants on the resulting transcripts. The resulting framework greatly improves on readily detecting clinically actionable structural variants.

Corresponding author
Miika J. Ahdesmäki,
miika.ahdesmaki@astrazeneca.com,
miika.ahdesmaki@live.fi

## INTRODUCTION

Structural variants (SVs) such as inversions, tandem duplications, large deletions and more complex chromosomal rearrangements are implicated as driver events in multiple cancers (*Latysheva & Babu, 2016*). Clinical detection of SVs in Mendelian diseases has been considered by e.g., *Noll et al. (2016)* but to our knowledge no prioritisation approach for oncology is publicly available. The mechanisms for oncogenic driver generation include activating fusions combining the coding frames (quite often in the intronic regions) of two genes, as well as truncating mutations in tumor suppressor genes or whole exon losses. Some well understood examples include TMPRSS2-ERG in prostate cancer (*Tomlins et al., 2008*), FGFR1,3-TACC1,3 in bladder and other cancers (*The Cancer Genome Atlas Research Network, 2014*), EGFRv3 deletion in glioblastoma and other tumours (*Sugawa et al., 1990*) and EML4-ALK in lung cancer (*Soda et al., 2007*).

The accurate calling of these complex, structural variants in short read DNA sequencing data is complicated by noise, manifested as false positives and lack of specificity. In many cases, the number of whole genome SV calls, including complex breakends, can be in the tens of thousands. While long read sequencing is likely to improve the calling of structural variants especially in germline DNA, tumour DNA from formalin-fixed paraffin-embedded (FFPE) tissue and circulating tumour DNA samples is inherently limited to short DNA fragment size. Utilising the currently available large amount of short read sequencing data to the full is therefore well motivated. It is also imperative to promptly pinpoint any clinically important structural variants when present in data.

In this paper we propose a tiered prioritisation approach to extract structural variants most likely to contribute to cancer proliferation and enable validation and follow up for a subset of high priority events. The prioritisation is based on greatly improved structural variant annotation in the variant annotation tool SnpEff (*Cingolani et al., 2012*) and can be applied to the output of any state of the art SV caller. Similar prioritisation work has been published in the domain of small variants, see for example *Carr et al. (2016)* and *Munz et al. (2015)*. The important aspect of easy and automated visualisation of the effects of structural variants on genes and coding exons is often overlooked with focus on structural variant calling algorithm performance. We thus further implement interactive structural variant visualisations in the New Genome Browser (https://github.com/epam/NGB). We show the full utility of the improved prioritisation and visualisation approaches in samples with structural variants leading to oncogenic gene fusions. The calling and filtering of RNA-seq based expressed, typically gain of function, fusions is well established and could be used to complement the DNA-focus of our approach; for example, see https://github.com/ndaniel/fusioncatcher (*Nicorici et al., 2014*).

## METHODS

The prioritisation approach proposed here bins SVs into tiers on predicted effect and builds on the output of any readily available SV calling algorithm. No new SV calling algorithm is proposed. In brief, most short read SV calling pipelines start with alignment of the DNA data to the human reference using an aligner like bwa-mem (*Li, 2013*). This is followed by the chosen SV caller integrating evidence from split and discordant reads, and potentially coverage (*Alkan, Coe & Eichler, 2011*), to make structural variant calls for deletions (DEL), tandem duplications (DUP), inversions (INV) and other more complex variants (BND). An example of these events is visualised in Fig. 1 in *Tattini, D'Aurizio & Magi (2015)*. The prioritisation is indifferent to the type of SV and it is the expected effect, such as a gene fusion that is of primary focus.

For SV calling we utilised two freely available SV callers that integrate evidence from split and discordant reads, Manta (*Chen et al., 2016*) and Lumpy (*Layer et al., 2014*). Both benchmark well in synthetic somatic data sets (see ICGC-TCGA DREAM Mutation Calling challenge leaderboards; https://www.synapse.org/#!Synapse:syn312572/wiki/247695) as well as germline reference standards (Genome in a Bottle NA12878). Any structural variant caller (such as BRASS; https://github.com/cancerit/BRASS) producing vcf files

compliant with the vcf specification (https://samtools.github.io/hts-specs/VCFv4.3.pdf) and compatible with SnpEff could equally well be used with our proposed methodology, provided they also quantify the numbers of discordant and split reads supporting at least the alternative allele.

To practically facilitate the prioritisation we improved annotations in SnpEff 4.3 to ease interpretation of fusion events, adding the Sequence Ontology (*Eilbeck et al., 2005*) annotation type *gene_fusion* for events where the open reading frames are in the same direction. Further, *bidirectional_gene_fusion* was introduced for where the frames of the putatively fused genes are opposing and therefore unlikely to be functional and *frameshift_variant* when the coding of the resulting fusion is out of frame, thus likely resulting in a truncated protein. The last two types are very important and interesting for loss of function of e.g., tumour suppressors. Other annotation improvements in SnpEff 4.3 include: *chromosome_number_variation*, *duplication* and *inversion*, which refer to large chromosomal deletions, duplications and inversions respectively (involving a whole exon, transcript, gene or even larger genomic regions), *exon_loss_variant* (whole or significant part of the exon was deleted) and *feature_ablation* (whole gene deleted).

We built a three tier prioritisation system (https://github.com/AstraZeneca-NGS/simple_sv_annotation, *Ahdesmaki (2016)*) using fusion and exon loss annotations. Given a list of genes of interest (GOI) we assign priorities, given in parentheses, as follows:

- Gene fusion
    - Fusion affecting two genes based on SnpEff annotation
        - on list of known pairs from FusionCatcher (*Nicorici et al., 2014*) (1)
        - one or both genes on GOI list (2)
        - neither on known pairs or GOI list (3)
    - Fusion affecting one gene
        - on GOI list (2)
        - not on GOI list (3)
- Whole exon loss based on SnpEff annotation
    - affecting a gene on GOI list (2)
    - not affecting a gene on GOI list (3)
- Upstream or downstream of GOI list genes based on SnpEff annotation (3)
- Other variant (REJECT)
- Missing ANN or SVTYPE in variant call file (REJECT)

The process is visualised in Fig. 1.

An example of a priority one gene fusion is given below, where the ANN field is the annotation provided by SnpEff. The duplication is interpreted by SnpEff as a gene fusion affecting FGFR3 and TACC3. This is a gene fusion on the list of known and published fusions and therefore is given priority one.

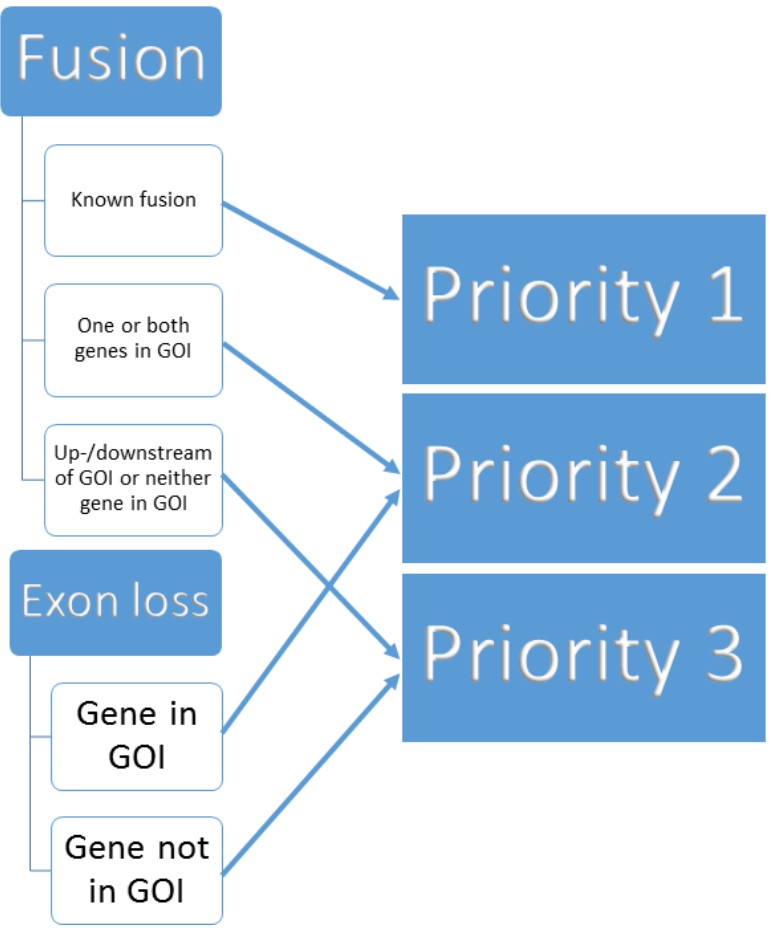

**Figure 1** Binning of structural variants into 3 priorities.

chr 4 1727831947 N <DUP> 0.0 PASS SVTYPE=DUP; SVLEN = 79207; END
= 1807038; STRANDS = −+:9072; CIPOS = 0,0; CIEND = 0,0; CIPOS95 = 0,0;
CIEND95 = 0,0; SU = 9072; PE = 4054; SR = 5018; AC = 0; AN = 0; ANN=<DUP>|
gene_fusion|HIGH|FGFR3&TACC3|ENSG00000068078&
ENSG00000013810|gene_variant|ENSG00000013810||||||||||GT:
SU:PE:SR:GQ:SQ:GL:DP:RO:AO:QR:QA:RS:AS:RP:AP:AB
./.:9072:4054:5018:.:.:−13017,−1748,−453:11177:3218:
7958:3217:7957:1516:4654:1701:3303:0.71

For the GOI, as a supplement to the prioritisation implementation we have provided a list of 300+ genes commonly associated with cancer, including genes involved in the MAPK and PI3K pathways (including receptor tyrosine kinase genes), DNA damage response, immuno-oncology and others. Alternatively, the user can provide their own gene lists in the implementation. The proposed prioritisation approach can be applied to variants from both paired (tumour/normal) and tumour only data, depending only on the structural variant callers' capabilities to handle paired samples. We confirmed the approach using TCGA data with known gene fusions.
**Table 1  Collection of structural variants leading to oncogenic fusions in different sample types.** All events are ranked into the highest category (1) by the prioritisation scheme.

| Sample | Panel or WGS | Manta call | Lumpy call | Fusion(s) |
|---|---|---|---|---|
| HDC134P rep. 1 | Panel with intronic probes | INV | INV | EML4-ALK |
| HDC134P rep. 2 | Panel with intronic probes | INV | INV | EML4-ALK |
| HDC134P rep. 3 | Panel with intronic probes | INV | BND | EML4-ALK |
| HDC140P rep. 1 | Panel with intronic probes | INV | BND | CCDC6-RET |
| HDC140P rep. 2 | Panel with intronic probes | INV | BND | CCDC6-RET |
| HDC140P rep. 3 | Panel with intronic probes | INV | BND | CCDC6-RET |
| HDC141P rep. 1 | Panel with intronic probes | BND | BND | ROS1-SLC34A2 |
| HDC141P rep. 2 | Panel with intronic probes | BND | BND | ROS1-SLC34A2 |
| HDC141P rep. 3 | Panel with intronic probes | BND | BND | ROS1-SLC34A2 |
| MCF7 | WGS | DUP | DUP | ESR1-CCDC170 |
| RT4 | Panel with intronic probes | DUP | DUP | TACC3-FGFR3 |
| PDX model | WES | DUP | DUP | TACC3-FGFR3 |
| Prostate cancer patient sample | FMI Panel with intronic probes | DEL | DEL | TMPRSS2-ERG |

# RESULTS AND DISCUSSION

## Prioritisation efficiently identifies clinically relevant gene fusions

We estimated the ability of our prioritisation approach to retain known mutations while reducing false positive events using samples with known structural variants (in a background of less well characterised SVs). Synthetic datasets from the ICGC-TCGA DREAM Mutation Calling challenge have also known artificial structural variants spiked in. These artificial SVs are useful, but do not however necessarily represent a realistic quantity of somatic SVs since they are not generated from a biological model, so we focused on known events in real sequenced samples. We collected sequencing data for seven samples with known SVs from cell lines, a patient derived xenograft and a clinical sample (Table 1). Although whole genome sequencing data or targeted capture including introns is preferred, any hybrid capture data can yield meaningful results if the breakpoints are close to captured regions or there are off target reads.

Following bwa-mem alignment to hg38, Lumpy and Manta both call the breakpoints and event types for the structural variants in Table 1, with slightly different interpretations for some like CCDC6-RET. As part of the updates to SnpEff we ensured that all these different types of SV events (INV, DUP, DEL) affecting two genes were correctly annotated as gene fusions.

The total number of calls for the samples in Table 1 as well as the numbers of variants falling into the tiers are shown in Table 2. The percentages of events falling into tier 1 ranged from between 0.2% (Manta MCF7 calls) to 40% (Manta HDC134P rep. 1). Focusing on the whole genome sequenced (WGS) MCF7 sample, the concordance of tier 1 events between Manta and Lumpy was fourteen SVs, with Manta having two private tandem duplications and Lumpy one private tandem duplication in addition to six private BND events. This followed the general trend of Lumpy calling more BND events than Manta. The read

**Table 2 Raw SV call numbers for Manta and Lumpy are given in the DUP, DEL, INV and BND columns.** The prioritised calls are shown in the last three columns. The Primary priority column corresponds to the number of detected fusions reported previously in the literature. All samples are from small hybrid capture panels except for the MCF7 sample, thus the relatively low numbers of calls per sample.

| Sample | Algorithm | DUP, DEL, INV | BND | Primary priority | Secondary priority | Tertiary priority |
|---|---|---|---|---|---|---|
| HDC134P rep. 1 | Manta | 5 | 0 | 2 | 2 | 0 |
| | Lumpy | 43 | 9 | 1 | 0 | 1 |
| HDC134P rep. 2 | Manta | 2 | 0 | 2 | 0 | 0 |
| | Lumpy | 41 | 4 | 1 | 0 | 0 |
| HDC134P rep. 3 | Manta | 3 | 0 | 2 | 0 | 0 |
| | Lumpy | 41 | 2 | 1 | 2 | 1 |
| HDC140P rep. 1 | Manta | 1 | 1 | 1 | 1 | 0 |
| | Lumpy | 48 | 20 | 1 | 2 | 0 |
| HDC140P rep. 2 | Manta | 1 | 1 | 1 | 1 | 0 |
| | Lumpy | 30 | 12 | 1 | 1 | 0 |
| HDC140P rep. 3 | Manta | 1 | 1 | 1 | 1 | 0 |
| | Lumpy | 56 | 9 | 1 | 3 | 0 |
| HDC141P rep. 1 | Manta | 0 | 2 | 2 | 0 | 0 |
| | Lumpy | 24 | 7 | 1 | 0 | 0 |
| HDC141P rep. 2 | Manta | 0 | 2 | 2 | 0 | 0 |
| | Lumpy | 17 | 1 | 1 | 0 | 0 |
| HDC141P rep. 3 | Manta | 0 | 2 | 2 | 0 | 0 |
| | Lumpy | 25 | 6 | 1 | 1 | 1 |
| MCF7 (WGS) | Manta | 8,239 | 1,990 | 16 | 48 | 2,814 |
| | Lumpy | 4,277 | 2,750 | 21 | 38 | 1,683 |
| RT4 | Manta | 169 | 158 | 1 | 20 | 135 |
| | Lumpy | 1,509 | 14,659 | 1 | 302 | 10,300 |
| PDX model | Manta | 248 | 65 | 5 | 2 | 164 |
| | Lumpy | 143 | 862 | 5 | 8 | 292 |
| Patient sample | Manta | 30 | 51 | 1 | 6 | 37 |
| | Lumpy | 1,034 | 3,621 | 1 | 13 | 177 |

support for the private SVs did not differ in any great manner from the shared SVs and therefore could still be true positives.

For MCF7 we provide a text file of the tier 1 and tier 2 events at https://github.com/AstraZeneca-NGS/publication_data, in addition to providing the vcf files.

In Table 2, the primary priority (known fusions) column lists the one true fusion known to be present in the HDC, RT4 cell lines and the patient sample. Manta reported two close but different breakpoints for the EML4-ALK fusion in the HDC134P replicates. The background of the PDX model is not fully characterised and therefore in the list of 5 fusions there may be false positives or SVs of unknown significance besides the FGFR3-TACC3 fusion. MCF7 has been previously characterised to at least contain the activating ESR1-CCDC170 fusion (*Veeraraghavan et al., 2014*) and be ESR1 driven. The remaining >10 fusions may be false positives or of less importance. This highlights that for the thousands of SV calls the prioritisation correctly draws the attention to the true positive events in the the primary priority. If no events of interest are found in the primary priority

bin the secondary priority can contain novel fusions or exon loss events of high interest as well. The tertiary priority (upstream, downstream events in genes of interest and fusions in genes of uninterest) is a catch-all category that should receive less attention.

To visualise the prioritised SV calls in the three replicate samples (HDC134P, HDC140P, HDC141P) run in triplicates, we utilised the UpSet package (*Lex et al., 2014*) to show con- and discordance in the prioritised calls. The plots in Figs. 2A–2C show the concordance histograms for each of the triplicates. The known fusions are detected by both Lumpy and Manta in all triplicates. There is additionally one structural variant (RET fused with chromosome 13) in HDC140P detected by all the algorithms in all the replicates. All the rest of the calls are private to one caller and one replicate (noise) but the number is small. This shows that the true events are very confidently called by both algorithms but there is a varying amount of false positives with Lumpy producing slightly more.

To show that the proposed approach correctly identifies the true events also in data not part of the sample set in Table 1, we applied the prioritisation to the TCGA bladder cancer cohort (*The Cancer Genome Atlas Research Network, 2014*). The FGFR3-TACC3 fusion in the RT4 cell line and patient derived xenograft (PDX) model in Table 1 is found in several TCGA samples, e.g., TCGA-CF-A3MF, TCGA-CF-A3MG and TCGA-CF-A3MH. Typically the breakpoints of gene fusions in the intronic regions, however in two of the TCGA samples (TCGA-CF-A3MH, TCGA-CF-A3MF), the FGFR3 breakpoint is in the last exon. The events were correctly annotated in these samples by our approach.

## Visualising gene fusions resulting from structural variants

Visualisation of structural variants to highlight the breakpoints and affected exons in a putative fusion transcript is an area of active development with no one tool currently being the industry standard. We initially identified Svviz (*Spies et al., 2015*), one of the earlier tools, to examine the validated fusion variants highlighted by prioritisation. The FGFR3-TACC3 tandem duplication (RT4 cell line) is shown in Fig. 3; TACC3 is not captured by the panel used and therefore no reads in support of the reference allele for TACC3 are shown. Svviz reassembles the reads around the putative breakpoints in its analysis and requires an amount of manual intervention.

We next decided to implement a variant call based gene fusion visualisation scheme in the open source New Genome Browser (NGB, https://github.com/epam/NGB). NGB takes the variant breakpoints annotated by SnpEff and uses Ensembl and UniProt based annotation to visualise the fusion product in both reference as well as the actual sequence context. The resulting plots highlight the fused exons of the affected genes. NGB uses vcf files as input and allows filtering based on vcf properties, including based on the priority from the prioritisation scheme to whittle down the putative variants.

The NGB visualisation of the FGFR3-TACC3 fusion is shown in Fig. 4A. Unlike the Svviz plot (Fig. 3), the visualisation is fully interactive html5 in the browser. Red highlighting is used to show the breakpoints relative to the coding regions in the alternative allele view and the red line shows the fusion points in the reference allele view. As NGB is a full feature genome browser, viewing both the read evidence as well as the fusion effects is simple. Figure 4B shows the read level evidence side-by-side from the two breakpoints.

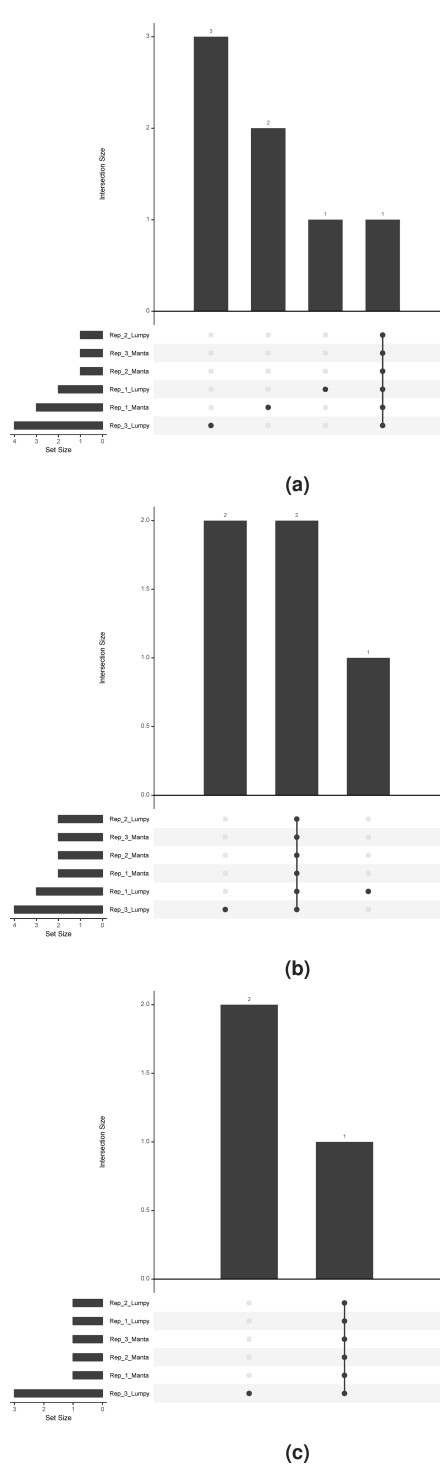

**Figure 2** **Prioritised SV call concordance.** The true positives are concordantly detected in addition to private (non-replicable) false positives. (A), (B) and (C) correspond to HDC134P (EML4-ALK), HDC140P (CCDC6-RET and RET-chr13), and HDC141P (SLC34A2-ROS1), respectively.
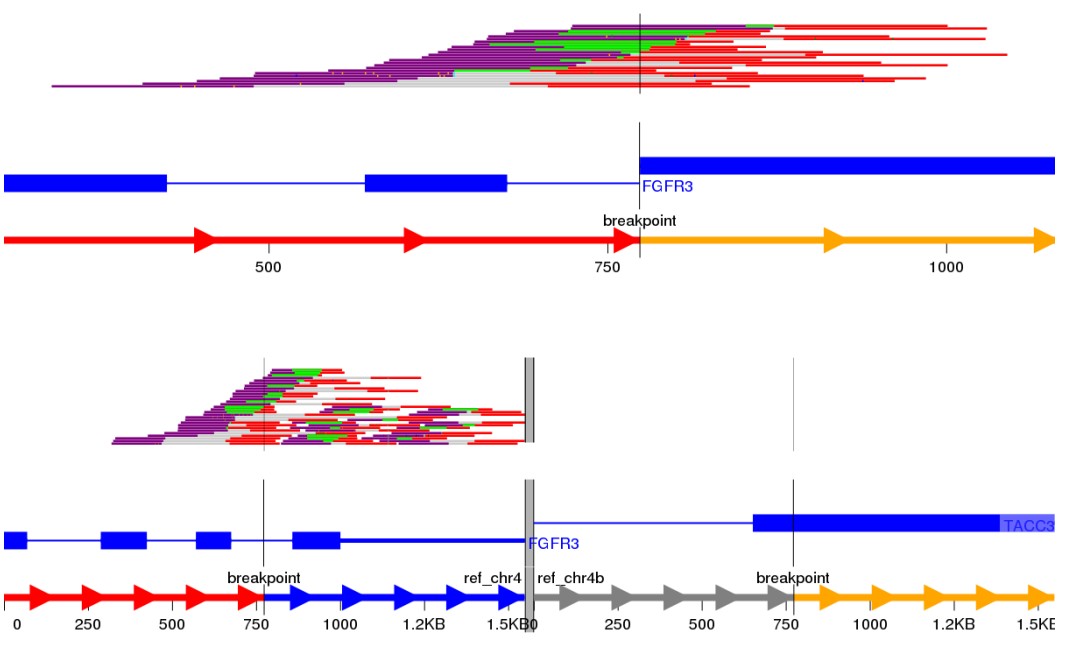

**Figure 3  Svviz output for the FGFR3-TACC3 fusion (tandem duplication) in the RT4 cell line.** Read evidence is shown for both how the last intron of FGFR3 is fused to an exon of TACC3 as well as for the reference alleles.

Soft clipping of the reads around the breakpoint are shown by the coloured base tails of the reads.

In Fig. 5 an interchromosomal translocation resulting in a fusion between ROS1 and SLC34A2 is shown. If multiple genes are overlapping the breakpoints NGB allows choosing the most relevant gene for the researcher. Another example for the EML4-ALK fusion that results from an inversion is shown if Fig. 6.

Other programs in development to better visualise the read level data include for example Genome Ribbon, see http://genomeribbon.com/.

## CONCLUSION

Here we presented a highly effective scheme for structural variant calling algorithms to prioritise for known fusion events as well as aberrations in a panel of cancer related genes.

This method prioritises based on biological information such as genes of interest and can be used in combination with orthogonal discovery based approaches (*Ganel et al., 2016*). In their approach, Ganel et al. produce in silico SV impact predictions that can be useful when whittling down the number of SVs of unknown significance and narrowing down to the likely most pathogenic ones; if the more hypothesis driven prioritisation described here does not yield satisfactory results, the approach by Ganel et al. might uncover additional novel variants or be used to provide a relative scoring for within-tier SVs. Future work also includes further incorporating protein domain information and which domains are retained in the suspected gene fusion.

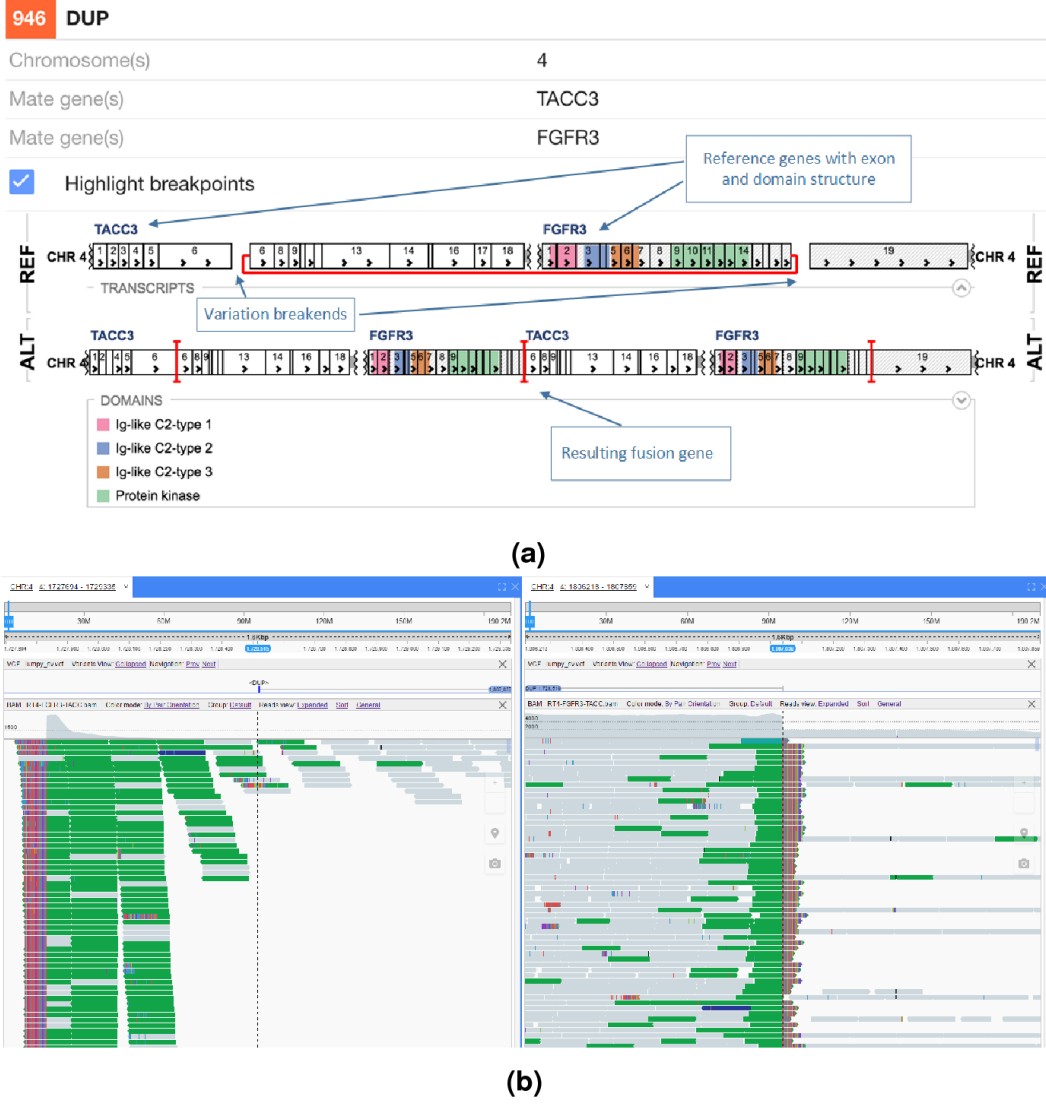

**Figure 4  FGFR3-TACC3 tandem duplication fusion exon level visualisation in the New Genome Browser.** Protein domains and exons affected by the structural variant are highlighted in colours. (A) shows the effect of the fusion and (B) the read evidence for the event at both breakpoints.

We further developed a visualisation framework in the New Genome Browser to illustrate the effects of the structural variants on genes in a user friendly, simple manner. We expect these visualisations to be extremely helpful for scientists in quickly producing publication ready gene fusion figures.

We look forward to suggestions from other groups to further improve structural variant calling interpretation and visualisation in cancer. This could be in the form of providing lists of genes of interest, suggesting alternative tiers to the prioritisation or adding support for other structural variant callers. The approach could also readily be extended to other fusion driven diseases by replacing the gene lists accordingly.

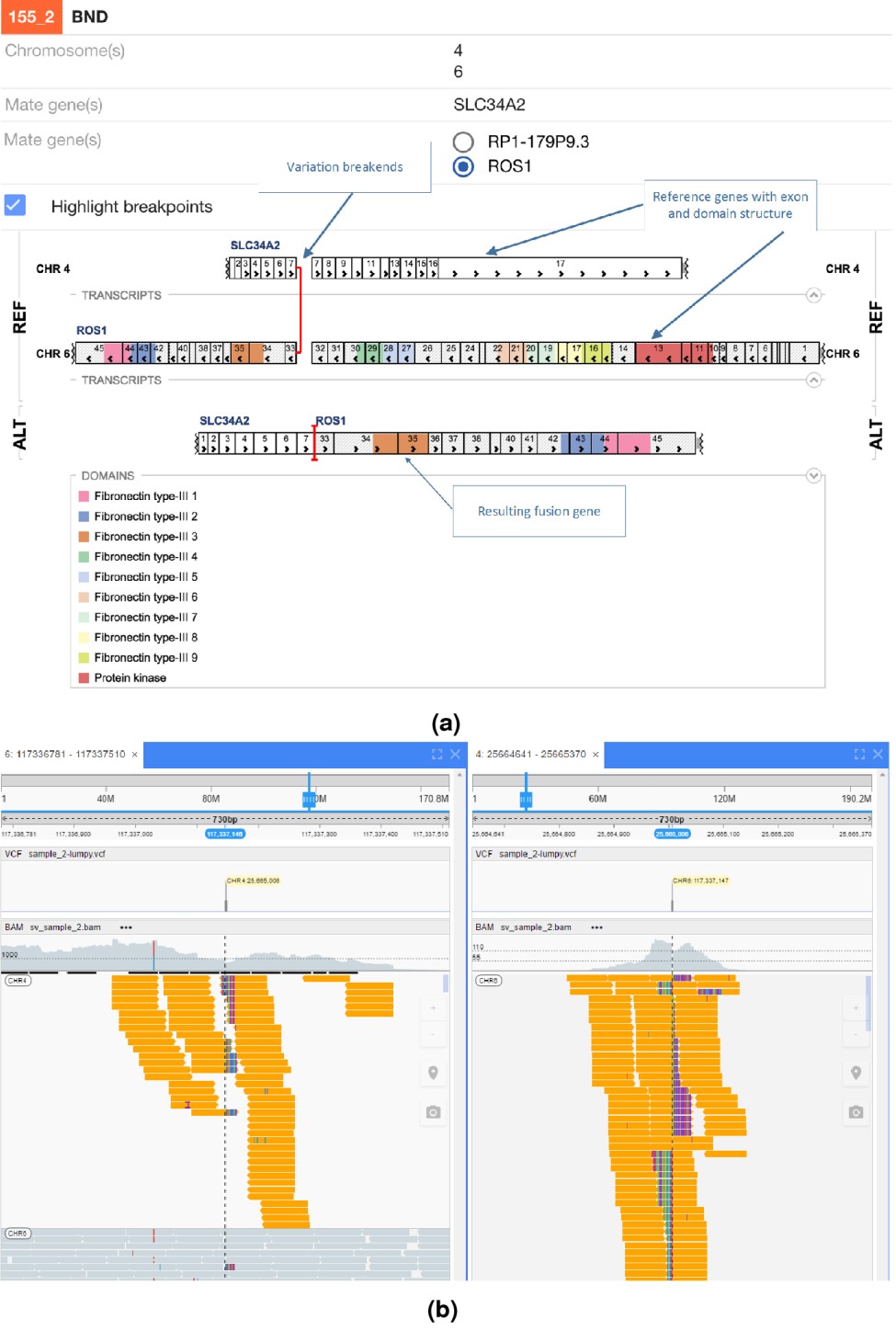

**Figure 5 ROS1-SLC34A2 interchromosomal translocation fusion.** (A) shows the effect of the fusion and (B) the read evidence for the event at both breakpoints.

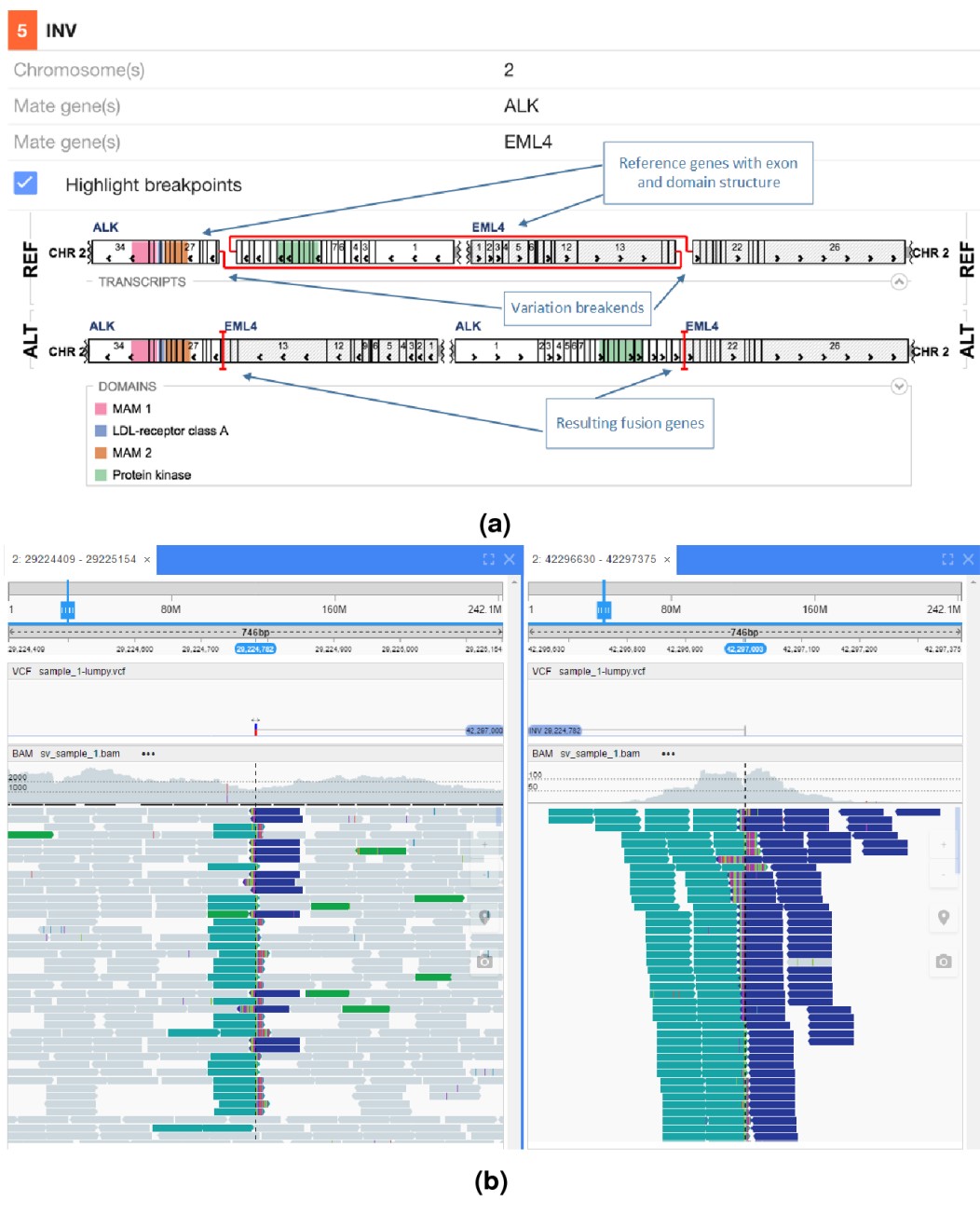

**Figure 6** **EML4-ALK inversion fusion.** (A) shows the effect of the fusion and (B) the read evidence for the event at both breakpoints.

## AVAILABILITY OF DATA AND ALGORITHMS

The structural variant call (vcf) level data will be made available for all the samples except for the PDX model at https://github.com/AstraZeneca-NGS/publication_data. The proposed prioritisation framework is fully integrated along with all the tools required to produce the results from raw sequencing data in bcbio (https://github.com/chapmanb/bcbio-nextgen). SnpEff 4.3 and later are available at http://snpeff.sourceforge.net/. The prioritisation code

for structural variants is accessible at https://github.com/AstraZeneca-NGS/simple_sv_annotation. The New Genome Browser is available at https://github.com/epam/NGB. All software used herein is freely available under open source licences.

## ACKNOWLEDGEMENTS

The following centres are kindly acknowledged for sequencing the three replicates of the samples HDC134P, HDC140P and HDC141P.

- West Midlands Regional Genetics Laboratory, Birmingham Women's NHS Foundation Trust, Mindelsohn Way, Edgbaston, Birmingham B15 2TG, UK, http://www.bwnft.nhs.uk/healthcare-professional/west-midlands-regional-genetics-laboratory
- Molecular Diagnostics, The Centre for Molecular Pathology, The Royal Marsden, 15 Cotswold Road, Sutton, Surrey SM2 5NG, UK, http://www.icr.ac.uk
- All Wales Medical Genetics Service, University Hospital of Wales, Heath Park, Cardiff CF14 4XW, UK

### Funding

The authors received no funding for this work.

### Competing Interests

Miika J. Ahdesmäki, Zhongwu Lai, T. Hedley Carr, Daniel Stetson, Brian Dougherty, J. Carl Barrett and Justin H. Johnson are employees of AstraZeneca Plc. Aleksandr Sidoruk, Gennadii Zakharov, Mikhail Rodichenko and Mikhail Alperovich are employees of EPAM Systems Inc. Pablo Cingolani is an employee of Kew Inc.

### Author Contributions

- Miika J. Ahdesmäki conceived and designed the experiments, analyzed the data, wrote the paper, prepared figures and/or tables, reviewed drafts of the paper.
- Brad A. Chapman and Pablo Cingolani analyzed the data, wrote the paper, reviewed drafts of the paper.
- Oliver Hofmann and J. Carl Barrett conceived and designed the experiments, wrote the paper, reviewed drafts of the paper.
- Aleksandr Sidoruk analyzed the data, prepared figures and/or tables, reviewed drafts of the paper.
- Zhongwu Lai and David Jenkins analyzed the data.
- Gennadii Zakharov, Mikhail Rodichenko and Mikhail Alperovich analyzed the data, prepared figures and/or tables.
- T. Hedley Carr performed the experiments, reviewed drafts of the paper.
- Daniel Stetson performed the experiments.
- Brian Dougherty and Justin H. Johnson conceived and designed the experiments, reviewed drafts of the paper.

## Data Availability

Vcf files available at https://github.com/AstraZeneca-NGS/publication_data.

Integration of proposed approach available at https://github.com/chapmanb/bcbio-nextgen.

Implementation of proposed structural variant prioritisation available at https://github.com/AstraZeneca-NGS/simple_sv_annotation.

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
