# Peer review of "Prioritisation of structural variant calls in cancer genomes"

_PeerJ, doi:10.7717/peerj.3166_

## Round 0.1 · original submission · Major Revisions

The manuscript has now been seen by 3 reviewers. They all find the work interesting but there needs to be more detail on the methods and comparison of the tool to other packages. Reviewer 2 commented that NGB is not fully setup on Github, please make sure the code and/or the executable are available.

Reviewer 1 ·

Basic reporting

The text is easy to follow in general, and references are for the most part adequate. However, there are some parts of the text that feels more like an internal documentation rather than a scientific paper. The methods section should be re-written almost from scratch.

For example, the third paragraph of the methods section ends with an incomplete sentence (Given a list of genes of interest (GOI)) -- where is the verb here? The following list gives the information used for prioritization, but both this sentence and the list is written very poorly. Also what are the numbers in parentheses in that list? Likely it refers to the three tiers, but this is not explained at all.

Another example of "internal documentation" is the mentioning of "bcbio" right after the list above as if it is a commonly known platform/toolkit. It is not, and it should be explained.

Minor comment: the sentence "Clinical detection of SVs in Mendelian diseases has been considered by others, see Noll et al. (2016)." at the end of the first paragraph in Introduction doesn't sound like it belongs there. All of that paragraph (and the paper) is devoted to cancer genomics, and mentioning Mendelian diseases sound weird. It may be moved maybe to an earlier position in the paragraph (maybe even as a second sentence), which can then be tied to lack of cancer SV prioritization.

Experimental design

The authors present a tool which loads, annotates and prioritizes variants loaded using standard VCF format. Experiments are basically analysis of available data sets and comparing with what is known in the literature, which are appropriate for this study.

Validity of the findings

As described above, findings are compared with the literature. Additional predictions still warrant further scrutiny and validation, however, not in the scope of this paper.

Additional comments

The tool sounds more like a classifier for SVs into three tiers, rather than finer scale prioritization. Some sort of scoring mechanism would be more useful to prioritize within tiers.

Reviewer 2 ·

Basic reporting

The manuscript is clearly written and understandable.
I would have liked to see more explanation of NGB in the method section. How does it work, what is it doing in the background...

I would suggest to merge Figure 1,2 and 3 since they basically show the same for different variants and in general do not hold much information. I could envision to replace them with some interesting findings that you see with your NGB.

Just to clarify: SVviz aims to validate SVs by realigning the reads over the reference and alternative region. It is not meant to be used to look at gene fusion per se.

Experimental design

The authors of the manuscript propose their method to:
1. Reduce false calls in the SVs data set
2. Priorities calls given previously known calls/gene fusions
3. Visualize the the gene fusion on the gene model.

i think 2 and 3 are nice ideas and I agree with what they suggest. I dont get the point how their method should reduce fasle calls. They or the user can of course use some consensus calling from other methods, but that is not part of this manuscript.

Plus as mentioned above the method section focuses only on the curation of the list of gene fusions or variants of interest. Since I think the novelty is the NGB it would be great to read more about this.

Validity of the findings

no comment

Additional comments

The authors of the manuscript propose a new visualization method focusing on gene fusion of cancer samples. I would suggest to provide more insights in the method that you have developed. I could envision that some figures to compare known gene fusions to prob. false fusions would be interesting.

You are often mentioning that your method reduces false calls, but I dont understand how. You cite other methods e.g. UpSet package, but this seems not to be included by default in your NGB.

Furthermore, NGB seems not to be available on Github. The repository is initialized with a ReadMe and License but no code or executable is provided.

Reviewer 3 ·

Basic reporting

Dear Editor,
Thank you for considering me as a reviewer for this paper.
in the paper, authors presented a prioritisation approach to pririotise known fusion events as well as aberrations in a panel of cancer related genes. Their prioritisation is based on structural variant annotation in the variant annotation tool SnpEff (a published tools by one of co-authors). they further developed a visualisation framework in a new genome browser to indicate the effects of the SVs on genes, which may be useful for other scientists in quickly producing publication ready gene fusion figures.

Overall, authors designed an integrative bioinformatics pipeline, which use several previously published tools for prioritisation. paper is a bioinformatics paper and has no biological interpretation; therefore, as a bioinformatics paper, it should has a clear structure for method section. However the method section is very brief and their main contribution in the pipeline is not clear. some improvements need to be done on method and introduction section.

Experimental design

null

Validity of the findings

null

Additional comments

1) what is different between current methods to identify SVs? please explain in more details in introduction section.
2) providing a flow-diagram about the proposed prioritisation approach will help readers better understand the paper.
3) what is the point of focusing on "DNA data only" for prioritisation of the SV Calls? what previous methods have done and what you are doing as a novelty; please explain in more detail.
4) the method section is very brief; I could not understand which part of their pipeline is new! it would be very helpful if authors provide the pipeline`s structure and highlight their main contribution in the pipeline.

---

## Round 0.2 · Minor Revisions

As you can see the reviewer's are happy with the revisions. Reviewer 1 and 2 have some minor items for you to fix before we can accept.

Reviewer 1 ·

Basic reporting

I have no further questions.

Experimental design

The authors present a tool which loads, annotates and prioritizes variants loaded using standard VCF format. Experiments are basically analysis of available data sets and comparing with what is known in the literature, which are appropriate for this study.

Validity of the findings

As described above, findings are compared with the literature

Additional comments

Minor: Galen et al paper is now published online in Bioinformatics. Please update references.

Reviewer 2 ·

Basic reporting

The manuscript is clearly written and is easy to understand and follow.

Experimental design

The authors described their prioritization approach in the methods. However, the manuscript lacks information on the gene fusion part of NGB.

E.g. how do you detect the gene fusions? Is it a greedy approach to say any SV connecting two genes? Do you allow for temporary hops to other regions?

Validity of the findings

All fine.

Additional comments

I really like the NGB gene fusion visualization. I would suggest that you also report the number of supported reads in the visualization. Since you already require this information it would help users to understand if this gene fusion is well supported or not.

Reviewer 3 ·

Basic reporting

it is ok now.

Experimental design

-

Validity of the findings

-

Additional comments

-

---

## Round 0.3 · accepted · Accept

Congratulations and thank you for responding to the reviewer's concerns.

---

## Author Rebuttal · Round 0.3

# ROUND 2

## Reviewer 1

### Basic reporting
I have no further questions.

### Experimental design
The authors present a tool which loads, annotates and prioritizes variants loaded using standard VCF format. Experiments are basically analysis of available data sets and comparing with what is known in the literature, which are appropriate for this study.

### Validity of the findings
As described above, findings are compared with the literature

### Comments for the author
Minor: Galen et al paper is now published online in Bioinformatics. Please update references.

## Our response

Dear Reviewer 1,
We have updated the reference accordingly, many thanks.

## Reviewer 2

### Basic reporting
The manuscript is clearly written and is easy to understand and follow.

### Experimental design
The authors described their prioritization approach in the methods. However, the manuscript lacks information on the gene fusion part of NGB.

E.g. how do you detect the gene fusions? Is it a greedy approach to say any SV connecting two genes? Do you allow for temporary hops to other regions?

### Validity of the findings
All fine.

**Comments for the author**

I really like the NGB gene fusion visualization. I would suggest that you also report the number of supported reads in the visualization. Since you already require this information it would help users to understand if this gene fusion is well supported or not.

## Our response

Dear Reviewer 2,

Just to clarify, NGB does not perform any fusion prediction or detection but merely reads information in the vcf file (mainly the SnpEff annotation). The text "We ... implement a variant call based gene fusion visualisation scheme in the open source New Genome Browser ... NGB takes the variant breakpoints **annotated by SnpEff** and uses Ensembl and UniProt based annotation to visualise the fusion product in both reference as well as the actual sequence context" ought to reflect this but if it is not clear enough please let us know. At the moment only simple events are considered where the breakpoints affect two genes (and are considered a fusion by SnpEff). Without long reads and phasing it would be very difficult to predict the effects of separate SVs affecting the same allele of a gene (i.e. hopping). We welcome suggestions on how to practically facilitate this on https://github.com/epam/NGB

Regarding the number of supported reads in the visualisation, there is already support for showing all vcf fields in the variants list and filter/sort by them. I am attaching a screenshot here showing "PE" (paired end reads) and "SR" (split reads), which were both selected from the "three horizontal bars" pop up menu top right:

| Type | | Chr | | Gene | Position | Pe | | Sr | | Info |
|------|---|-----|---|------|----------|-----|---|-----|---|------|
| INV | | 2 | | EML4, A… | 29224782 | 413 | | 0 | | i |
| DEL | | 3 | | PIK3CA | 1792199… | 5 | | 0 | | i |
| BND | | 5 | | BTNL9 | 1810501… | 6 | | 6 | | i |
| BND | | 5 | | BTNL9 | 1810502… | 6 | | 6 | | i |
| BND | | 6 | | | 51295112 | 5 | | 0 | | i |
| BND | | 6 | | | 51295400 | 5 | | 0 | | i |
| BND | | 6 | | ROS1 | 1173147… | 4 | | 0 | | i |

We have recently modified the SV-simplification program to provide an INFO field of the highest priority of all possible effects of a single variant to help sort the variants better in NGB. For any further queries please open tickets on Github.